# Memory-Efficient Learning of Stable Linear Dynamical Systems for Prediction and Control

**Giorgos Mamakoukas**
Northwestern University
giorgosmamakoukas@u.northwestern.edu

**Orest Xherija**
University of Chicago
orest.xherija@uchicago.edu

**Todd Murphey**
Northwestern University
t-murphey@northwestern.edu

## Abstract

Learning a stable Linear Dynamical System (LDS) from data involves creating models that both minimize reconstruction error and enforce stability of the learned representation. We propose a novel algorithm for learning stable LDSs. Using a recent characterization of stable matrices, we present an optimization method that ensures stability at every step and iteratively improves the reconstruction error using gradient directions derived in this paper. When applied to LDSs with inputs, our approach—in contrast to current methods for learning stable LDSs—updates both the state and control matrices, expanding the solution space and allowing for models with lower reconstruction error. We apply our algorithm in simulations and experiments to a variety of problems, including learning dynamic textures from image sequences and controlling a robotic manipulator. Compared to existing approaches, our proposed method achieves an *orders-of-magnitude* improvement in reconstruction error and superior results in terms of control performance. In addition, it is *provably* more memory efficient, with an $\mathcal{O}(n^2)$ space complexity compared to $\mathcal{O}(n^4)$ of competing alternatives, thus scaling to higher-dimensional systems when the other methods fail. The code of the proposed algorithm and animations of the results can be found at `https://github.com/giorgosmamakoukas/MemoryEfficientStableLDS`.

## 1   Introduction

Linear dynamical systems arise in many areas of machine learning and time series modeling with active research applications in computer vision [2], robotics [28], and control [8, 19, 20]. Linear representations are often desirable because they admit closed-form solutions, simplify modeling, and are general enough to be useful in many applications (e.g. Kalman filters). Further, there are well-established tools for the analysis (e.g. investigating properties of a system, such as stability and dissipativity), prediction, estimation, and control of linear systems [16]. They are, in general, computationally more efficient than nonlinear systems and highly promising candidates for real-time applications or data-intensive tasks. Last but not least, linear dynamical models can also be used to capture nonlinear systems using Koopman operators, which linearly evolve nonlinear functions of the states [22, 4, 27, 15].

LDSs are models that are learned in a self-supervised manner and are therefore promising for data-driven applications. Consequently, with the availability of higher computational power and the wide applicability of data-driven modeling, there is renewed interest in learning LDSs from data. Examples include learning spatio-temporal data for dynamic texture classification [2, 10],

video action recognition [24, 37], robotic tactile sensing [25] and nonlinear control using Koopman operators [4, 3]. Although linear system identification is a well-studied subject [26, 29], algorithms that learn LDSs from data have often overlooked important properties, such as stability.

Stability describes the long-term behavior of a system and is critical both for numerical computations to converge and to accurately represent the true properties of many physical systems. When stability is overlooked, the learned model may be unstable even when the underlying dynamics are stable [7], in which case the long-term prediction accuracy dramatically suffers. This is why there are increasing efforts to impose stability on data-driven models [2, 18, 21, 11, 1]. However, the available methods do not scale well or are not applicable for control.

In this work, we present a novel method for learning stable LDSs for prediction and control. Using a recent characterization of matrix stability [14], we derive a gradient-descent algorithm that iteratively improves the reconstruction error of a projected stable model. Contrary to current top-performing methods that start from the least-squares (LS) solution and iteratively push the LDSs towards the stability region, our method enforces stability in each step. As a result, it returns a stable LDS even after one single iteration. This feature can become crucial in online applications and time-sensitive tasks where obtaining a stable state-transition matrix as early in the optimization process as possible becomes of central importance. Furthermore, whereas alternative methods terminate upon reaching stability, our method can iterate on already stable solutions to improve the reconstruction error. It can therefore be used to further improve the solutions of other methods.

Our proposed method is *provably* more memory efficient, with an $\mathcal{O}(n^2)$ space complexity—$n$ being the state dimension—compared to $\mathcal{O}(n^4)$ of the competing alternative schemes for stable LDS. For systems with inputs, we derive the gradient directions that update both state and control linear matrices. By doing so, we expand the space of possible solutions and enable the discovery of models achieving lower error metrics compared to searching only for a stable state matrix which, to the best of our knowledge, is what the current top-performing algorithms do.

To demonstrate the superior performance of our method, we test it on the task of learning dynamic textures from videos (using benchmark datasets that have been used to assess models that learn stable LDSs), as well as learning and controlling (in simulation and experiment) the Franka Emika Panda robotic arm [12]. When compared to the current top-performing models, a constraint generation (CG) [2] and a weighted least squares (WLS) [18] approach, our method achieves an orders-of-magnitude lower reconstruction error, robustness even in low-resource settings, and better control performance. Notably, our approach is the first that tests the control performance of stable LDS; CG has been formulated but not evaluated for control tasks and it is not straightforward that WLS can be implemented for such applications, as the results in this paper suggest.

The paper is structured as follows. In Section II, we review linear systems and stability. In Section III, we introduce and derive the proposed algorithm for learning stable LDSs. In Section IV, we compare our method to competing alternative algorithms that learn stable LDSs in prediction and control. In Section V, we discuss our findings and point to areas for future research.

## 2 Linear Dynamical Systems

We consider states $x \in \mathbb{R}^N$, controls $u \in \mathbb{R}^M$ and discrete time LDSs modeled as

$$y_t \equiv x_{t+1} = Ax_t + Bu_t, \tag{1}$$

where $A \in \mathbb{R}^{N \times N}$ and $B \in \mathbb{R}^{N \times M}$ are the state and control matrices, respectively. For systems without inputs, one can simply set $B = 0$. We use $\mathcal{S}_{A,B} = \{(A, B) \mid x_{t+1} = Ax_t + Bu_t\}$ to denote the solution space of the matrices $A$ and $B$ that describe a LDS of the form (1). Further, let $\{\lambda_i(A)\}_{i=1}^N$ be the eigenvalues of an $N \times N$ matrix A in decreasing order of magnitude, $\rho(A) \equiv |\lambda_1(A)|$ be the spectral radius of $A$, and $\mathbb{S}$ be the set of all stable matrices of size $N \times N$.

### 2.1 Learning Data-Driven LDSs

Next, we provide an overview of data-driven learning of LDSs. First, we consider systems without control for which CG and WLS were developed. Later, in Section 3, we modify the learning objective to include control terms and learn stable representations for LDSs with inputs.

Given $p$ pairs of measurements $(x_t, y_t)$, learning LDSs from data typically takes the form

$$\hat{A} = \inf_A \frac{1}{2}\|Y - AX\|_F^2, \tag{2}$$

where $Y = [y_1\, y_2\, \ldots\, y_p] \in \mathbb{R}^{N \times p}$, $X = [x_1\, x_2\, \ldots\, x_p] \in \mathbb{R}^{N \times p}$, and $\|\cdot\|_F$ is the Frobenius norm. The LS solution is then computed as

$$A_{ls} = YX^{\dagger}. \tag{3}$$

where $X^{\dagger}$ denotes the Moore-Penrose inverse of $X$. The optimization problem in (2) does not impose stability constraints on $\hat{A}$. To learn stable LDSs, the learning objective is typically formulated as

$$\hat{A} = \inf_{A \in \mathbb{S}} \frac{1}{2}\|Y - AX\|_F^2, \tag{4}$$

and is highly nonconvex.

The current top-performing methods for computing stable LDSs are a constraint generation [2] and a weighted least squares [18] approach. CG formulates the optimization as a quadratic program without constraints, which is an approximation to the original problem. It then iterates on the solution to the approximate optimization by adding constraints and terminates when a stable solution is reached. WLS determines the components of the LS transition matrix that cause instability and uses a weight matrix to enforce stability, while minimizing the reconstruction error. Note that both methods consider an entire sequence of observations, say $\mathcal{D} \in \mathbb{R}^{N \times p}$, such that $X = \mathcal{D}_{[0:p-1]}$ and $Y = \mathcal{D}_{[1:p]}$, thereby making the assumption that all measurements belong to a unique time-series dataset. In the case of the WLS method, this assumption is necessary and the method fails dramatically for datasets with disjoint windows of time, as we demonstrate later in Section 4.3. CG and our proposed method, on the other hand, do not require contiguous observations.

## 2.2   Subspace Methods

For high-dimensional LDSs, as is the case with image reconstruction, it is computationally prohibitive to learn a state transition matrix. Even for small images of size $100 \times 100$ pixels, the dimensionality of the state transition matrix $A$ would be $100^4$. For such high-dimensional systems, models are obtained using subspace methods that reduce the dimensionality of the learning task. Subspace methods for learning LDSs typically apply singular value decomposition (SVD) on the original dataset [17] decomposing the observation matrix $\mathcal{D} \approx \mathcal{U}\Sigma V^T$, where $\mathcal{U} \in \mathbb{R}^{N \times r}$, $V \in \mathbb{R}^{p \times r}$ are orthonormal matrices, $\Sigma = \{\sigma_1, \ldots, \sigma_r\} \in \mathbb{R}^{r \times r}$ contains the $r$ largest singular values, and $r < N$ is the subspace dimension. Then, the learning optimization is performed on the reduced observation matrix $D_r = \Sigma V^T$, with $X_r = D_{r[0:p-1]}$ and $Y_r = D_{r[1:p]}$. $\mathcal{U}$ is used to project the solutions back to the original state space. For a more complete description of standard subspace methods, the reader can refer to [6, 30, 33, 36, 35].

## 3   The Algorithm

The optimization problem for finding stable LDSs has traditionally only considered solving for a stable matrix $A$ that minimizes the reconstruction loss. In this work, we formulate the objective as

$$[\hat{A}, \hat{B}] = \inf_{A \in \mathbb{S}, B} \frac{1}{2}\|Y - AX - BU\|_F^2, \tag{5}$$

to expand the solution space and solve both for a stable state matrix $A$ and a matrix $B$. We denote the least-square solution for the control system by $[A_{ls}, B_{ls}] = Y \cdot [X; U]^{\dagger}$.

### 3.1   Optimization Objective and Gradient Descents

The proposed algorithm uses a recent characterization of stable matrices [14]. Specifically, a matrix $A$ is stable if and only if it can be written as $A = S^{-1}OCS$, where $S$ is invertible, $O$ is orthogonal, and $C$ is a positive semidefinite contraction (that is, $C$ is a positive semidefinite matrix with norm less than or equal to 1). By constraining the norm of $C$, one bounds the eigenvalues of $A$ and ensures stability. Using this property, we formulate the optimization problem as

$$[\hat{A}, \hat{B}] = \inf_{S \succ 0, O \text{ orthogonal}, C \succeq 0, \|C\| \leq 1} \frac{1}{2}\|Y - S^{-1}OCSX - BU\|_F^2, \tag{6}$$

where $\hat{A} \equiv S^{-1}OCS$. Then, for $f(S,O,C,B) = \frac{1}{2}\|Y - S^{-1}OCSX - BU\|_F^2$, we derive the gradient directions with respect to the four matrices $S, O, C$, and $B$ as follows:

$$\nabla_S f(S,O,C,B) = S^{-T}EX^TS^TC^TO^TS^{-T} - C^TO^TS^{-T}EX^T \tag{7}$$

$$\nabla_O f(S,O,C,B) = -S^{-T}EX^TS^TC^T \tag{8}$$

$$\nabla_C f(S,O,C,B) = -O^TS^{-T}EX^TS^T \tag{9}$$

$$\nabla_B f(S,O,C,B) = -EU^T \tag{10}$$

where $E = Y - S^{-1}OCSX - BU$. Due to space constraints, the derivation of the gradients is presented in the supplementary material. We then use the fast projected gradient descent optimization from [13] to reach a local minimum of the reconstruction cost. The algorithmic steps are presented in Algorithm 1. The proposed algorithm enforces stability in every iteration step by projecting the solution onto the feasible set. For more details, the reader can refer to [13] or the provided code.

Henceforth, we refer to our proposed algorithm as SOC. Note that, contrary to CG and WLS that search stable LDSs in $\mathcal{S}_{A,B_{l_s}}$ by iterating over only $A$, SOC updates both linear matrices $A$ and $B$, thereby expanding the feasible solution space to $\mathcal{S}_{A,B}$, where $\mathcal{S}_{A,B} \supset \mathcal{S}_{A,B_{l_s}}$. Further, SOC does not assume time continuity of the training measurements, contrary to WLS. The novelty of SOC with respect to [14] is the derivation of new gradient directions that not only account for control inputs, but that are also calculated so as to best fit training measurements instead of finding the nearest stable solution to an initial unstable matrix.

---

**Algorithm 1** SOC Algorithm using Fast Gradient Method (FGM) with restart from [13]

---

**Input:** $X, Y, U$                                                      ▷ State and control measurements
**Output:** $A \in \mathbb{S}, B$                                                           ▷ Stable LDS
1: Initialize $Z \triangleq (S,O,C,B), k_{max}, \gamma_o, \lambda \in (0,1), \alpha_1 \in (0,1)$
2: $\hat{Z} = Z$
3: **while** $k < k_{max}$ **do**
4:    $Z_k = \mathcal{P}(\hat{Z} - \gamma\nabla f(\hat{Z})); \ \gamma = \gamma_o$           ▷ $\mathcal{P}$ is the projection to the feasible set
5:    **while** $f(Z_k) > f(Z)$ **and** $\gamma \geq \gamma_{min}$ **do**        ▷ Line search to find gradient step size
6:       $Z_k = \mathcal{P}(\hat{Z} - \gamma\nabla f(\hat{Z}))$
7:       $\gamma = \lambda\gamma$
8:    **end while**
9:    **if** $\gamma < \gamma_{min}$ **then**                                  ▷ If line search fails, FGM restarts
10:       $\hat{Z} = Z; \ a_k = a_1$
11:    **else**                                      ▷ If cost is decreased, the solution is stored
12:       $\alpha_{k+1} = \frac{1}{2}(\sqrt{\alpha_k^4 + 4\alpha_k^2} - \alpha_k^2); \ \beta_k = \frac{\alpha_k(1-\alpha_k)}{\alpha_k^2 + \alpha_{k+1}}$
13:       $\hat{Z} = Z_k + \beta_k(Z_k - Z); \ Z = Z_k$
14:    **end if**
15: **end while**
16: $A = S^{-1}OCS$
17: **return** $A \in \mathbb{S}, B$

---

## 4 Experiments

We implement LS, CG, WLS, and the proposed SOC method for learning LDSs and compare their performance on dynamical systems with and without control inputs. We omit the seminal work of [23] in our comparisons as it has been outperformed in terms of error, scalability, and execution time by both CG and WLS. For systems without inputs, we focus on learning dynamic texture from frame sequences extracted from videos using standard benchmark datasets [32, 5, 31]. For systems with inputs, we use experimental data from the Franka Emika Panda robotic manipulator and illustrate the learning and control performance of all the methods considered. We split the results in three parts: memory requirements, reconstruction error performance, and control performance. For an impartial assessment, we perform all comparisons in MATLAB using the publicly available code of the CG

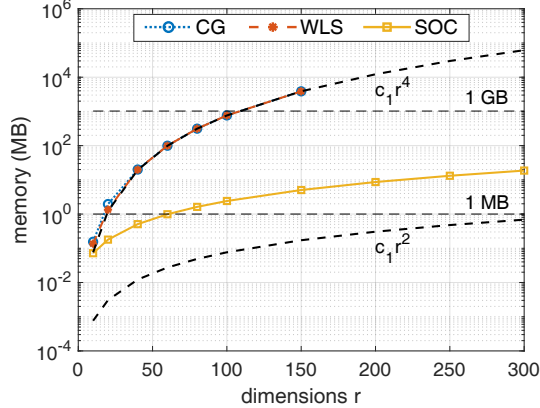

Figure 1: Memory usage as a function of dimensions $r$, where $c_1 = 8/2^{20}$. CG and WLS scale proportionately to $r^4$, whereas SOC scales proportionately to $r^2$. For $r = 150$, SOC uses about 5.04 MB, whereas CG and WLS about 3.78 GB. Due to memory limits, WLS and CG failed to run at higher dimensions.

and WLS algorithms[1]. All simulations are performed using MATLAB R2019b on a machine with a 14-core Intel E5-2680v4 2.4-GHz CPU with 20GB RAM.

## 4.1 Memory Usage

First, we compare the three algorithms on their memory demands. For an objective comparison, we only measure the size of all MATLAB workspace variables created by the algorithms. That is, we consider a matrix with 4 double-precision cells to use 32 bytes. We compare the algorithms on a sequence of frames extracted from a coffee cup video downloaded from Youtube[2]. We use this video because it exhibits dynamical motion and has a sufficient number of frames to allow for relatively higher subspace dimensions (the SVD decomposition limits the subspace dimension to be no larger than the number of frames).

The results are shown in Figure 1. SOC scales proportionately to $r^2$, whereas both CG and WLS scale proportionately to $r^4$. This is because CG and WLS both rely on solving a quadratic programming problem with a state dimension $n^2$, which generates matrices of dimension $n^4$, whereas SOC uses a gradient descent approach that employs only matrix inversion, transposition, multiplication and addition, all of which are operations of space complexity $O(n^2)$. At $r = 150$, SOC uses about 5.04 MB of memory; CG and WLS use about 3.78 GB of memory and fail to run at higher dimensions due to memory constraints. Though such high dimensions may perhaps seem out of scope for the image reconstruction examples demonstrated next, they can typically occur in the field of robotics. For example, a recent study [3] used a linear data-driven Koopman representation with dimensions $r = 330$ to identify and control a pneumatic soft robotic arm. For this dimension, WLS and CG would require about 88 GB of memory and SOC would need about 25 MB. As a result, only SOC would be able to successfully train a stable Koopman model on a standard personal laptop and, as we show in the control performance section, failing to impose stability on the learned model can lead to unsafe robot movements.

## 4.2 Error Performance

To measure the predictive accuracy of the learned representations, we use three benchmark datasets: UCLA [32], UCSD [5], and DynTex [31]. The UCLA dataset consists of 200 gray-scale frame sequences that demonstrate 50 different categories of dynamic motion (e.g. flame flickering, wave motion, flowers in the wind), each captured from 4 different viewpoints. Every frame sequence contains 75 frames of size $48 \times 48$ pixels. The UCSD dataset consists of 254 frame sequences showing highway traffic in different environmental conditions. Each sequence contains between $42$ and $52$ frames of size $48 \times 48$ pixels. For the DynTex dataset, we use 99 sequences from 5 groups of

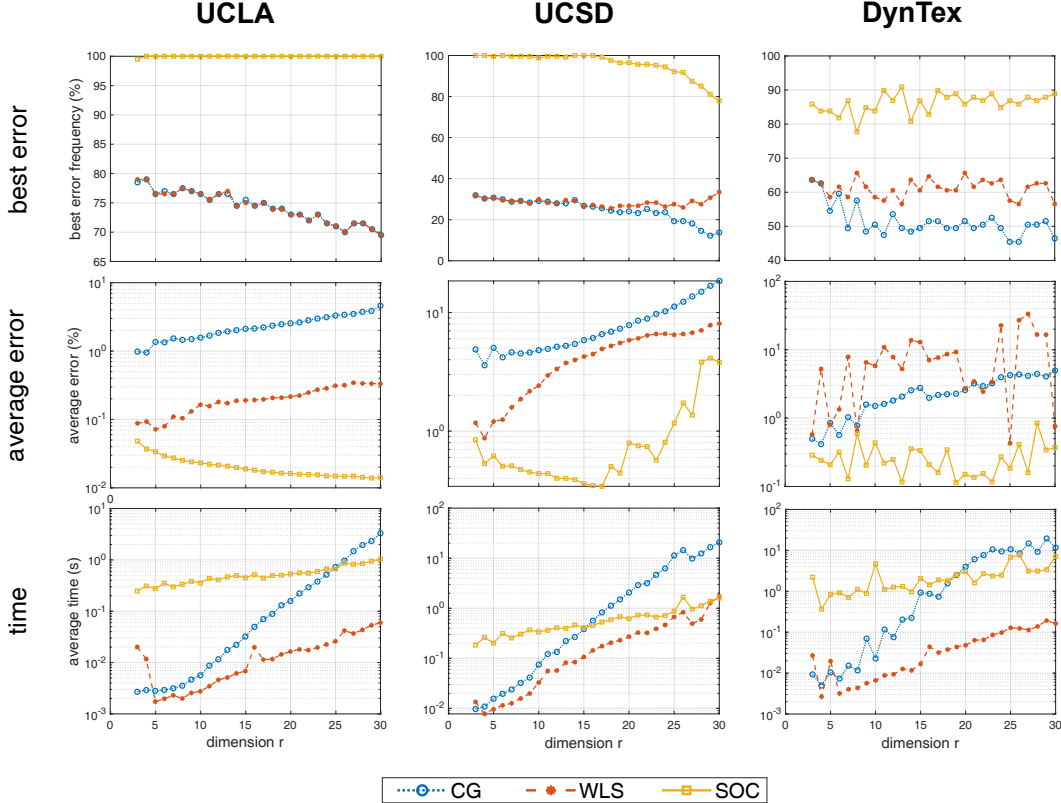

Figure 2: Learning performance of CG, WLS, and SOC for varying subspace dimensions performed on three datasets: UCLA, UCSD, and DynTex. In all cases, SOC has the highest best error frequency, has lower average error and, in terms of execution time, scales better than the other methods.

dynamic texture (`smoke` and `rotation` from the Beta subset and `foliage`, `escalator`, and `flags` from the Gamma subset) that exhibit periodic motion. The frames are of size $352 \times 288$ pixels. We convert the frames to grayscale and use the bicubic interpolation algorithm implemented in the Python library `pillow` to scale down the frames without ratio distortion down to $48 \times 39$ pixels. Each DynTex sequence contains between $250$ and $1576$ frames.

As explained in Section 2, the dimensionality of images can be prohibitively high and cause slow computations or memory failures: the transition matrix for an image of size as small as $48 \times 48$ pixels would require hundreds of TBs for CG and WLS to run. For this reason, we use subspace methods to reduce the problem dimensionality. For each dataset, we consider a set of subspace dimensions $r \in \{3, 30\}$. Then, for each dimension, we use the four methods (LS, CG, WLS, and SOC) to obtain a LDS for each of the frame sequences. To compare the performance of the four algorithms, we use the reconstruction error relative to the LS solution: $e(\hat{A}) = \frac{e(\hat{A}) - e(A_{ls})}{e(A_{ls})} \times 100$.

We report the results in Figure 2 and focus on three metrics: best error frequency, average reconstruction error, and execution time. The best error graphs plot the percentage of frame sequences for a given dimension for which an algorithm computes the best relative error (that is, lower than or equal to the other two methods). This metric credits all schemes that achieve the lowest error and so curves may add up to more than 100%. The average error and time graphs show the average reconstruction error and average execution time of all frame sequences for each dimension, respectively.

Across the three datasets, SOC computes the best error for more frame sequences than the other methods across any dimension. In the UCLA and UCSD datasets, the SOC best error frequency reaches 100% for the majority of the dimensions contrary to less than 80% (for UCLA) and 40% (for UCSD) attained by CG and WLS. This means that, for the aforementioned datasets, CG and WLS only rarely find a better solution than SOC. While for the DynTex dataset the differences are not as pronounced, SOC still computes the best error for most of the frame sequences for any dimension

Table 1: Performance on the `steam`, `fountain`, and `coffee cup` sequences. Results that did not converge to a stable solution are indicated with −.

| | SOC | CG | WLS | SOC | CG | WLS | SOC | CG | WLS |
|---|---|---|---|---|---|---|---|---|---|
| | (r = 20) | | | (r = 40) | | | (r = 80) | | |
| **steam** | | | | | | | | | |
| $\|\lambda_1\|$ | 1 | 1 | 1 | 1 | 1 | 1 | 1 | 1 | 1 |
| $\sigma_1$ | 1.06 | 1.03 | 1.07 | 1.10 | 1.03 | 1.10 | 2.03 | 1.05 | 4.64 |
| $e(\hat{A})$ | **12.32** | 28.05 | 24.94 | **5.59** | 24.90 | 21.27 | **6.38** | 25.21 | 10.98 |
| time (s) | 0.36 | **0.22** | 0.53 | **1.48** | 9.76 | 15.82 | 5.45 | 1146.23 | 456.11 |
| **fountain** | | | | | | | | | |
| $\|\lambda_1\|$ | 1 | 1 | 1 | 1 | 1 | 1 | 1 | - | - |
| $\sigma_1$ | 1.11 | 1.00 | 1.11 | 1.43 | 1.01 | 1.43 | 1.04 | - | - |
| $e(\hat{A})$ | **0.001** | 1.07 | 0.004 | **0.0005** | 2.97 | 0.0007 | **169.38** | | - |
| time | 1.52 | 0.48 | **0.18** | 3.18 | 15.40 | **0.84** | 5.96 | - | 63.85 |
| **coffee cup** | | | | | | | | | |
| $\|\lambda_1\|$ | 1 | 1 | 1 | 1 | 1 | 1 | 1 | 1 | 1 |
| $\sigma_1$ | 1.02 | 1 | 1.21 | 1.04 | 1.01 | 1.79 | 1.11 | 1.07 | 1.09 |
| $e(\hat{A})$ | **2.85** | 6.20 | 321.21 | **3.30** | 5.84 | 562.17 | **0.36** | 1.08 | 2.14 |
| time | **0.27** | 30.71 | 0.44 | **0.95** | 2.99 | 9.28 | 38.30 | 24.25 | **6.44** |

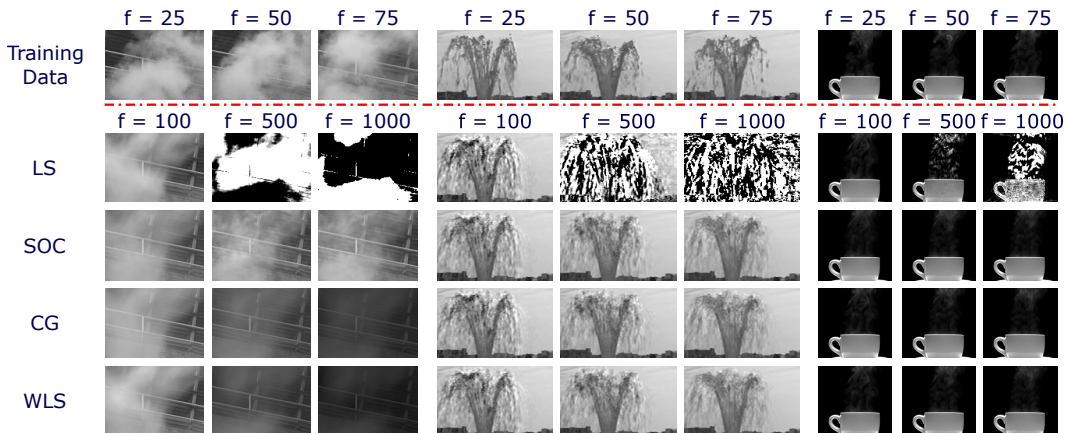

Figure 3: Synthesized sequences generated by LS, SOC, CG, and WLS for $r = 40$. Supporting videos can be found at `https://github.com/giorgosmamakoukas/MemoryEfficientStableLDS`.

and about 20% more often than the other methods. Second, SOC has orders-of-magnitude lower average relative error across all dimensions and datasets. Last, in terms of the execution time, SOC is slower than CG and WLS for low dimensions ($r < 20$). However, it scales better than the other two methods, such that it becomes faster than CG for $r > 20$. For the UCSD dataset, SOC and WLS become comparable in terms of average execution time near $n = 30$. This observation is in line with the fact that CG and WLS are high space-complexity algorithms that may even fail to perform at high dimensions due to memory limitations.

Next, we compare the three methods on the `steam` sequence (composed of $120 \times 170$ pixel images) and the `fountain` sequence (composed of $150 \times 90$ pixel images) from the MIT temporal texture database [34], together with the `coffee cup` sequence used in Figure 1. Results are shown in Table 1. To show the effect on the predictive quality of the solutions, we plot the frames reconstructed from the learned LDS for each method in Figure 3. Note that the LS solution degrades over time and generates unrealistic frames.

## 4.3 Control

In this section, we demonstrate the superior performance of our approach in control systems. Using experimental data gathered from the robotic arm Franka Emika Panda, we illustrate the improvement in both the reconstruction error of the learned model and the control performance. To use CG and

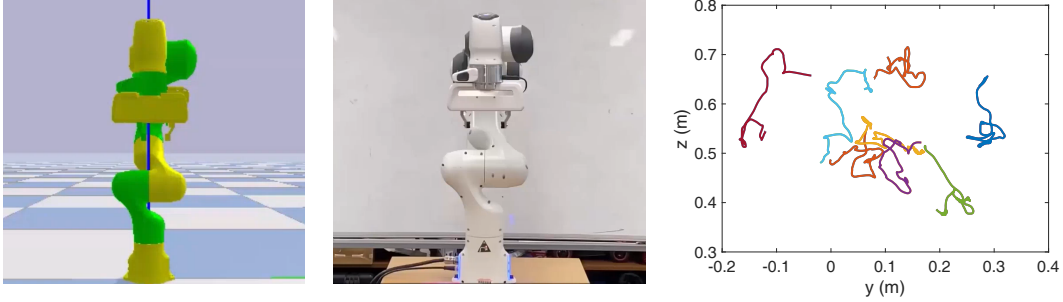

Figure 4: From left to right: simulation environment, physical robot, and experimental training data.

Table 2: Errors of stable LDS using experimental data from the Franka Emika Panda manipulator.

| Measurements | 50 | 75 | 100 | 150 | 200 | 300 | 500 |
|---|---|---|---|---|---|---|---|
| SOC | **0.01** | **0.56** | **0.0001** | **0.06** | **0.05** | **0.09** | **0.05** |
| CG | 50.14 | 32.77 | 11.66 | - | - | 1.40 | 0.23 |
| WLS | 17.00 | - | - | 124.01 | 36.63 | 25.17 | 42.01 |

WLS to compute a stable $\hat{A}$, we use the LS solution for the control matrix and modify the objective to

$$\hat{A} = \inf_{A \in \mathbb{S}} \frac{1}{2} \|Y' - AX\|_F^2, \tag{11}$$

where $Y' = Y - B_{ls}U$. The learning performance is then measured as the % error increase when compared to the LS solution $(A_{ls}, B_{ls})$. Note that this error depends both on $\hat{A}$ and $\hat{B}$; for WLS and CG, we use the LS solution for the control matrix $(B = B_{ls})$, whereas SOC computes both $A$ and $B$.

We collected training data on the experimental platform at 50 Hz, using a controller to manually move the robotic arm. We gathered 400 measurements (8 seconds) in eight separate runs. The training data, along with the experimental and simulation environments used in this section are shown in Figure 4. Table 2 compares the performance of the SOC, CG, and WLS algorithms on learning stable models for the Franka Emika Panda robotic manipulator using experimental data. The performance is compared for different numbers of measurements $p$. As the data show, SOC is the only algorithm that never fails to find stable solutions, regardless of the amount of training data. As more measurements are used, the LS solution itself becomes more stable and CG and WLS are both able to converge to stable solutions. Further, the quality of CG solutions improves with more training measurements; the performance of SOC remains robust throughout the testing cases.

In Figure 5, we plot the reconstruction error for the three methods for different training data sizes. In this setting, however, measurement sets $(x_t, y_t, u_t)$ are randomly drawn from the training data such that the matrices $Y$ and $X$ have discontiguous measurements. Note how such a choice worsens the performance of WLS that assumes continuity in the observation matrices. On the other hand, CG and SOC are similar in learning performance.

With regard to controlling the system, we use LQR control computed using the models from each algorithm and simulate tracking a figure-8 pattern. The states are the $x, y, z$ coordinates of the end effector, the 7 joint angles of the arm, and the 7 joint angular velocities and the applied control is the joint velocities. The trajectory is generated in the $y - z$ plane for the end effector; the desired angle configurations of the robotic arm are solved offline using inverse kinematics; the desired angular joint velocities are set to 0. LQR control is generated using $Q = \text{diag}([c_i]) \in \mathbb{R}^{17 \times 17}$, where $c_i = 1$ for $i \in \{1, 10\}$ and 0 elsewhere and $R = 0.1 \times I_{7 \times 7}$.

The LS model is unstable and fails at the task. Similarly, WLS—despite the stable model—performs poorly, highlighting the need for both stability and fidelity of the learned representation. On the other hand, CG and SOC are similar in performance.

To measure robustness across the initial conditions, we run 50 trials, varying both the $y$ and $z$ initial positions with displacements sampled uniformly in $\mathcal{U}(-0.1, 0.1)$. Across all trials, LS has an average error of 7556, WLS scores 38.73, CG scores 0.0810 and SOC scores 0.0799.

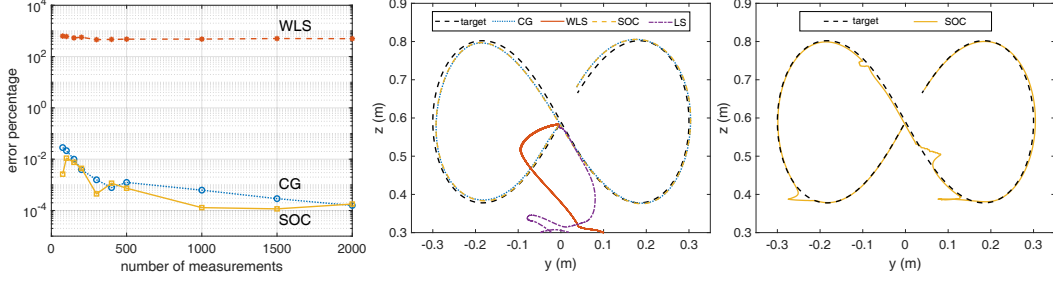

Figure 5: Control performance in simulation using experimental measurements from the Franka Emika Panda robotic arm. The left figure shows the reconstruction error of the learned models for a varying number of measurements sampled randomly from the training set; the middle figure shows the performance of the controllers after training with 100 measurements sampled randomly (2 seconds worth of data); the right figure shows the control performance of SOC after manually introducing disturbances to the position of the end effector.

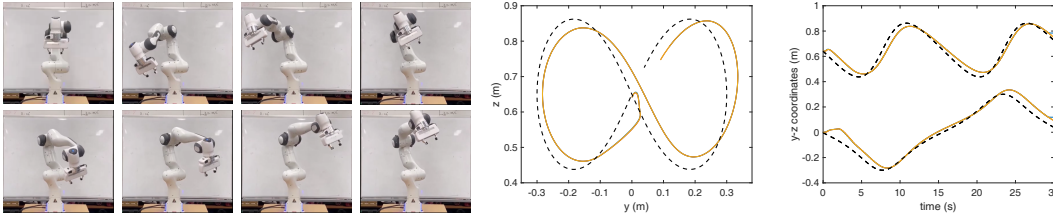

Figure 6: Experimental tracking of a figure-8 pattern using the Franka Emika Panda robotic manipulator. The left figure shows, from top to bottom, snapshots of the control maneuver; the rest figures show the trajectories of three trials. The three trials are almost identical, showing the robustness of the method. The applied control is computed with an LQR policy using the stable LDS system obtained from the SOC algorithm. The training data are obtained using 600 measurements.

Then, we test LQR control computed on the LDS obtained from the SOC algorithm in an experiment to demonstrate that the simulation results are indicative of the performance in a physical experiment. Figure 6 shows the control performance of three trials tracking a figure-8 pattern. Due to COVID-19 limitations, we were unable to extend the experimental tests. However, these results serve primarily to experimentally validate our approach and illustrate that the simulation results are an accurate prediction of the experimental behavior as well.

## 5   Conclusion

In this work, we introduce a novel algorithm for computing stable LDSs. Compared to the current top-performing alternatives, the proposed scheme is significantly more memory efficient and, as a result, scales better for high-dimensional systems often encountered in image processing and robotic applications. Further, the suggested method outperforms the alternatives in terms of error and control performance, as demonstrated on three benchmark datasets and the Franka Emika Panda robotic arm experiments. These features make it a promising tool for compression and data-driven system identification tasks.

Coupled with the ongoing research around Koopman-operator-based nonlinear control, this algorithm can be a promising candidate for high-dimensional nonlinear control and other machine learning applications, as well. Indeed, recent work in [9] uses Koopman operators to optimize training of neural network methods; also work in [38] learns deep neural network models for Koopman operators of nonlinear dynamical systems. Imposing stability on Koopman operators represented using basis functions learned via deep learning will combine the benefits of linear representations with the predictive power of neural networks.

## Broader Impact

Our methods can improve robotic tasks that are safety-critical, particularly those that include a human-in-the-loop (such as rehabilitation devices and prosthetics) where the human-robot interaction dynamics are not known ahead of time. For such tasks, a robotic platform prioritizes stability and safety during operation. Unstable data-driven models may lead to catastrophic robotic behavior, as we demonstrate in our simulations with the Franka Emika Panda robotic arm. Our work provides a mechanism for online learning of models that satisfy stability constraints, improving the safety and reliability of closed-loop control of those systems.

## Acknowledgments and Disclosure of Funding

First and foremost, we thank Nicolas Gillis for the communication and useful discussions about the fast gradient method. We also thank Ian Abraham for his help with the experimental testing on the Franka Emika Panda robot and Wenbing Huang for very kindly providing us with the datasets and results used previously to test the WLS algorithm. We also thank the anonymous reviewers for their invaluable comments that helped improve the quality of this manuscript. Last, we gratefully acknowledge the Research Computing Center (RCC) of the University of Chicago for providing the computing resources to execute our experiments and simulations. This work is supported by the National Science Foundation (IIS-1717951). Any opinions, findings, and conclusions or recommendations expressed in this material are solely those of the author(s) and do not necessarily reflect the views of any of the funding agencies or organizations.

## Footnotes

[1] https://github.com/huangwb/LDS-toolbox

[2] https://www.youtube.com/watch?v=npkBC4GYodg

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
