[Supplementary Material]

# Supplementary Material: Memory-Efficient Learning of Stable Linear Dynamical Systems for Prediction and Control

**Giorgos Mamakoukas**
Northwestern University
giorgosmamakoukas@u.northwestern.edu

**Orest Xherija**
University of Chicago
orest.xherija@uchicago.edu

**Todd Murphey**
Northwestern University
t-murphey@northwestern.edu

## A Gradient Descents for SOC algorithm

Here we derive the gradient descents for the SOC algorithm, shown in equations (7) through (10) in the main paper. Let

$$f = \frac{1}{2}\|Y - S^{-1}OCSX - BU\|_F^2$$

Using the identity

$$\|X\|_F^2 = \text{Tr}(X^T X)$$

and expanding the product $(Y - S^{-1}OCSX - BU)^T(Y - S^{-1}OCSX - BU)$, we rewrite the optimization problem as

$$
\begin{aligned}
f =& \frac{1}{2}\text{Tr}(Y^T Y - Y^T S^{-1}OCSX - Y^T BU \\
& - X^T S^T C^T O^T S^{-T} Y + X^T S^T C^T O^T S^{-T} S^{-1}OCSX + X^T S^T C^T O^T S^{-T} BU \\
& - U^T B^T Y + U^T B^T S^{-1}OCSX + U^T B^T BU).
\end{aligned}
$$

To calculate the gradients of $f$, we use the identities

$$\frac{\partial}{\partial X}\text{Tr}(AXB) = A^T B^T \tag{12}$$

$$\frac{\partial}{\partial X}\text{Tr}(AX^T B) = BA \tag{13}$$

$$\frac{\partial}{\partial X}\text{Tr}(B^T X^T CXB) = (C^T + C)XBB^T. \tag{14}$$

The gradient of $f$ with respect to $C$ is

$$
\begin{aligned}
\frac{\partial}{\partial C}f =& \frac{1}{2}(-O^T S^{-T} YX^T S^T - O^T S^{-T} YX^T S^T + (2O^T S^{-T} S^{-1}O)CSXX^T S^T \\
& + O^T S^{-T} BUX^T S^T + O^T S^{-T} BUX^T S^T) \\
=& -O^T S^{-T}(Y - S^{-1}OCSX - BU)X^T S^T. \tag{15}
\end{aligned}
$$

Similarly, the gradient of $f$ with respect to $O$ is

$$
\begin{aligned}
\frac{\partial}{\partial O}f =& \frac{1}{2}(-S^{-T}YX^TS^TC^T - S^{-T}YX^TS^TC^T + (2S^{-T}S^{-1})OCSXX^TS^TC^T)\\
& + S^{-T}BUX^TS^TC^T + S^{-T}BUX^TS^TC^T)\\
=& -S^{-T}(Y - S^{-1}OCSX - BU)X^TS^TC^T.
\end{aligned}
\tag{16}
$$

The gradient of $f$ with respect to $B$ is

$$
\begin{aligned}
\frac{\partial}{\partial B}f =& \frac{1}{2}(-YU^T + S^{-1}OCSXU^T - YU^T + S^{-1}OCSXU^T + 2BUU^T)\\
=& -(Y - S^{-1}OCSX - BU)U^T.
\end{aligned}
\tag{17}
$$

Last, the gradient of $f$ with respect to $S$ is calculated as follows:

$$
\frac{1}{2}\|Y - S^{-1}OCSX - BU\|_F^2. = \langle R - Y | R - Y \rangle,
$$

where $R = S^{-1}OCSX + BU$. Then, using the property

$$
\frac{\partial X^{-1}}{\partial q} = -X^{-1}\frac{\partial X}{\partial q}X^{-1}
$$

we calculate

$$
\dot{R} = -S^{-1}\dot{S}S^{-1}OCSX + S^{-1}OC\dot{S}X,
$$

such that

$$
\begin{aligned}
\dot{f} =& \frac{1}{2}\langle \dot{R} | R - Y \rangle + \langle R - Y | \dot{R} \rangle\\
=& \langle R - Y | \dot{R} \rangle\\
=& \langle R - Y | -S^{-1}\dot{S}S^{-1}OCSX + S^{-1}OC\dot{S}X \rangle\\
=& \langle -S^{-T}(R - Y)X^TS^TC^TO^TS^{-T} + C^TO^TS^{-T}(R - Y)X^T | \dot{S} \rangle.
\end{aligned}
$$

Thus, the gradient of $f$ with respect to $S$ is

$$
\begin{aligned}
\frac{\partial}{\partial S}f =& S^{-T}(Y - S^{-1}OCSX - BU)X^TS^TC^TO^TS^{-T}\\
& - C^TO^TS^{-T}(Y - S^{-1}OCSX - BU)X^T.
\end{aligned}
\tag{18}
$$

To simplify the notations, we use $E = Y - S^{-1}OCSX - BU$ and rewrite the gradients (15) through (18) as

$$
\begin{aligned}
\nabla_C f =& -O^TS^{-T}EX^TS^T\\
\nabla_O f =& -S^{-T}EX^TS^TC^T\\
\nabla_B f =& -EU^T\\
\nabla_S f =& S^{-T}EX^TS^TC^TO^TS^{-T} - C^TO^TS^{-T}EX^T,
\end{aligned}
$$

where we use the notation $\nabla_X(\cdot) \equiv \frac{\partial}{\partial X}(\cdot)$.