[Reviews · NeurIPS 2020]

Review 1

Summary and Contributions: This paper leverages a novel characterization of the space of stable matrices into a novel algorithms for estimating the parameters of a linear dynamical system from data. The algorithm provides marked improvements to the estimation error and the memory complexity compared to standard approaches.

Strengths: This is a very clear paper, with a simple premise, and a derivation which explores both the theoretical and practical consequences of the new algorithm. The math is clearly stated and detailed. The multiple evaluation domains provide a strong degree of confidence in the value of the approach. The improvements appear material and relevant.

Weaknesses: Topically, this work is not directly about machine learning since it describes a pure optimization method. However, estimation of LDS is relevant to the overall problem of deriving control strategies for data, so this isn't necessarily an issue, though it may be more valued if published at a venue more centrally devoted to control and optimization. No mention is made as to whether the attached code will be open-sourced, which would greatly enhance the value of the paper.

Correctness: Yes

Clarity: Very clear and well organized.

Relation to Prior Work: Work builds on a recent result that was independently derived, but provides a novel way to leverage the work and valuable empirical characterizations. Reviewer is not familiar enough with the literature to comment on related prior art.

Reproducibility: Yes

Additional Feedback:


Review 2

Summary and Contributions: This paper proposed a new method to learn stable linear models for dynamic systems. It utilized a recent characterization of stable matrices and constructed the gradient descent method to improve the reconstruction error within the stable linear model subspace. The proposed method has O(n^2) space complexity and achieved reconstruction errors that were orders of magnitude better than the baselines.

Strengths: The proposed algorithm seems to be a significant improvement over the state-of-the-art: The space complexity is O(n^2) instead of O(n^4). The reconstruction error is orders of magnitude better than the baseline.

Weaknesses: Although the linear dynamic system is widely used because it is convenient in math, the majority of the dynamic systems in our world is nonlinear. This paper argues that the method can be used with the Koopman operator for nonlinear systems. However, no example is shown. Experiments of combining Koopman operator or Differential Dynamic Programming with the learned stable linear model would make the paper much stronger.

Correctness: The high level derivation seems good. I did not carefully check the detailed derivation in the supplemental material, though.

Clarity: In general, the paper is well written. It would be great to include accompanying videos to visualize the results. For example, comparing the reconstructed videos and the ground truth would be much more convincing than only showing a few frames in Figure 3. Similarly, videos of Franka robot tracking the desired patten would be helpful. The paper has a detailed derivation of gradients of the optimization (6). It is not immediately clear to me how to make sure that the matrices satisfy the constraints: S is positive definite, O is orthogonal, C is positive semi-definite and contraction. Some explanations would be helpful. I do not quite understand the best error frequency plot (Top row of Figure 2): The sum of the best error frequencies of the three methods does not add up to 100. For example, in the UCLA dataset, the best error frequency of SOC is 100%. Shouldn't the best error frequency of CG and WLD be 0% then? What is the control U in the Franka example? Is it desired joint angle, desired joint velocity, or motor torque?

Relation to Prior Work: The relation to prior work is clearly and sufficiently discussed.

Reproducibility: Yes

Additional Feedback: I have read the author's response. It addressed my questions and I will keep my positive score.


Review 3

Summary and Contributions: The paper proposes a new algorithm for learning stable linear dynamical systems. The algorithm is based on a recent characterization of stable matrices which shows that a matrix is stable if and only if it can be written as a product of certain psd and orthogonal matrices. The proposed algorithm uses O(n^2) space in contrast to O(n^4) space used by previous methods, where n is the state dimension. The authors show that the new algorithm gets lower reconstruction error compared to prior art on benchmark datasets. The paper also extends their approach to the control setting, where they jointly search over the state and control matrices. By doing this, they achieve superior performance on control tasks compared to an algorithm which uses prior work to estimate a stable state matrix and then does Least Squares to estimate the control matrix.

Strengths: The experimental evaluation is very thorough and quite convincing. Since this is mainly an experimental paper, I think this is probably the most important criterion for evaluating it. Within the experimental evaluation, the algorithm seems to do well on multiple fronts: 1) it uses much less memory, 2) it is more scalable as the dimensionality increases, 3) it gets lower reconstruction error, and 4) it does better on control tasks. The authors also evaluate the control task on a real robotic manipulator which shows the synthetic setup is an accurate representation of the real world task. The provided code in the supplementary is also very well-organized, kudos to the authors for doing a great job on this.

Weaknesses: One could argue that the algorithm is not very novel, given the previous characterization of stable matrices. But I don't think this is too much of a concern as the characterization has not been exploited for this purpose before, and the major contribution here is probably the empirical evaluation. I found that the paper could emphasize the importance of stability more. For instance, it is nice to note that the LS solution which does not enforce stability does get worse over time in Fig. 3. Since the utility of the algorithm rests on the premise that it is important for the linear system to be stable, I think this premise should be supported a bit more.

Correctness: The empirical methodology is very solid. I think it would be good to verify somewhere that the algorithm only needs O(n^2) space. On a related note, why is it that the prior algorithms needed O(n^4) space, if the matrices are only nxn?

Clarity: The paper is well written and is easy to read.

Relation to Prior Work: The authors mostly do a good job of discussing prior work. One thing that would be good to clarify is whether there was prior work in using stable matrices for the control task, since CG and WLS were formulated for the prediction task?

Reproducibility: Yes

Additional Feedback: 1. It would be interesting to examine the source of the better prediction accuracy of SCO. Are the solutions obtained by SCO more stable than CG and WLS? If not, what is the reason for the superior error performance? More generally, it could be good to more closely examine and discuss the results in Fig. 2, as some of the observed relations seem to demand further attention. For instance in the UCSD dataset, it seems that increasing the state dimension even hurts average error. Is this because the intrinsic dimension of the system is small, and estimating models with higher dimension would require more data? Also, for the UCLA dataset the average error of SCO reduces with dimension, whereas it increases for the other algorithms. 2. It would have been nice to evaluate the algorithm on a setup where the state dimension needs to be high, in for example robotics settings. This is because one of the key proposed advantages of SCO is that it is scalable. 3. I am curious how the prediction accuracy of the algorithms compare on the coffee cup sequence. A few additional remarks regarding presentation: 1. In Section 2.1, the authors shifts to systems without inputs somewhat abruptly, since in the beginning of Section 2 they considered the general case with inputs. 2. The authors should write CG and WLS in parenthesis the first time they use these abbreviations. ----Post author response---- I thank the authors for the detailed and helpful response. All of the points mentioned in the response are valid and help in understanding the work, and I encourage the authors to include it in the revision. I feel more confident about my score now, and have updated the confidence to reflect this.


Review 4

Summary and Contributions: This paper proposes a method to learn stable dynamical systems by an iterative optimisation method. The method preserves stability at every iteration update scheme. The proposed approach is evaluated in simulations to a variety of problems, including learning dynamic textures from image sequences and controlling a robotic manipulator. Compared to existing approaches, the proposed method achieves an orders-of-magnitude improvement in reconstruction error and superior results in terms of control performance

Strengths: Claims are sound and empirical evaluation well-conducted. The paper is well written, notation and claims are clear. Extensive experimentations and results help to understand the contribution of the paper. Authors clearly state their contributions and the gains they observed from their experimental results, both in term of performance scores and memory-efficiency. The proposed method compares favourably with respect to existing algorithms on certain datasets in terms of average error.

Weaknesses: The paper is at times ill-motivated. It is unclear why the dynamical systems estimated from the datasets considered should be stable. This is particularly true for the control experiments, with manipulation tasks leading inherently to unstable dynamics. The novelty with respect to existing system identification methods (such as stability preserving subspace identification methods and methods based on Riemannian optimisation) is unclear.

Correctness: Claims and methods are correct to the best of my understanding. The empirical methodology is correct. However, it is unclear why prediction methods are not considered.

Clarity: The paper is well written and well structured. Results are presented in a clear way and help to understand and highlight the contribution of the paper.

Relation to Prior Work: Literature review can be improved. Additional references concerning stability preserving subspace identification methods would strengthen the paper considerably.

Reproducibility: Yes

Additional Feedback: The code submitted is clear and well documented, favouring reproducibility and further research in this field.


Review 5

Summary and Contributions: This paper presents a novel algorithm to learn stable linear dynamical system from data. Based on the characterization of matrix stability method proposed by [22], they derive a gradient-descent algorithm to optimize the learned model to satisfy constraints by minimizing the constructed error. Extensive experiments including on common benchmarks and robot arm control, demonstrate that their proposed SOC outperforms existing important baselines.

Strengths: 1. I quite like the presentation of this paper, which is easy to understand. Their key idea, i.e. introduction of the characterization of matrix stability into learning model is demonstrated and evaluated well. 2. Learning stable LDSs is useful and relatively underexplored compared with other learning based control areas. Their proposed method is simple to implement and effective in ensuring the stability in the learning based LDSs. 3. Their SOC outperforms the competing baselines, CG and WLS by a large margin. The robot arm experiment they conducted is interesting, and can reflect effectiveness of the learned control model. 4. SOC also has the advantage of efficient memory, which is important in many robot applications. 5. Their code is provided. Overall, I think this is a good work that is useful in learning based control.

Weaknesses: 1. Can you briefly and intuitively introduce the reason why the characterization of stable matrices can ensure the stability of LDSs in your main paper? Although this is not your main contribution, it would be good for readers to better understand your idea, and more convincing, given a short introduction of this technique. 2. Can you talk about the assumption of the characterization of stable matrices and your derived gradient descent algorithm for it? It will help to understand the applicability of your method.

Correctness: Their claims and method are technically sounding.

Clarity: This paper is well-structured and easy to understand.

Relation to Prior Work: They have discussed the previous work in the introduction.

Reproducibility: Yes

Additional Feedback:

[Author Response · NeurIPS 2020]

We would like to thank the reviewers for the feedback they provided. Following is our response.

**Reviewer 1**: The attached code will be open-sourced and available on GitHub in Python and MATLAB. Besides control and robotic applications, our method is relevant to machine learning as it can be applied without change to Koopman operators, which are increasingly gaining attention in deep learning. For example, recent work by Dogra et al. (2020) uses Koopman operators to optimize training of neural network methods; also work by Yeung et al. (2019) learns deep neural network models for Koopman operators of nonlinear dynamical systems. Imposing stability on Koopman operators that are based on basis functions learned via deep learning could lead to a scheme that combines the benefits of linear representations with the predictive power of neural networks in data-driven prediction and control.

**Reviewer 2**: The proposed algorithm can be readily applied to Koopman operators. We will include an experimental example of a robot learning how to push a planar block—an open problem in robot learning—illustrating both the need for stable operators and the performance when one closes the feedback loop. We will include animations of the results in Fig. 3 and Fig. 6 with our code release. The constraints on $S$, $O$, and $C$ are enforced at every iterate. This is shown in the provided code. We will include the algorithmic steps and details about the projections in the revised manuscript and in the code repository. The 'best error frequency' metric credits all schemes that achieve the top performance such that curves may add up to more than 100% when two or more methods have obtained the same best error. The Franka control is the joint velocities.

**Reviewer 3**: With respect to the novelty of the algorithm, we use the characterization of the stable matrices to derive *new* gradient descent directions to compute matrices that are stable *and* minimize the reconstruction error with respect to data measurements. We will point out in the paper that the LS solution degrades over time and add supporting videos of the results in Fig. 3.

To our knowledge, this is the first work that tests the control performance of stable LDS. CG has been formulated but not evaluated for control tasks and is not straightforward that WLS can be implemented for such applications, as the results in our paper suggest. There is previous work by Ng et al. (2004) on rejecting unstable controllers in online learning, but the suggested method does not generate stable models and control. Instead it can only filter solutions, which would not be helpful if the implemented controller is always unstable. With respect to space complexity, the matrix operations and sizes in our algorithm require at most $O(n^2)$ memory. The CG and WLS algorithms use a quadratic programming solution with $n^2$ state dimension, which generates matrices of dimensions $n^4$ hence a space complexity of $O(n^4)$.

We attribute the difference in performance across the datasets to the type of motion represented in them. The UCLA and DynTex datasets contain periodic, stable motion that is more suitable for the algorithms examined; the UCSD dataset includes highway traffic with cars entering and exiting the frame, manifesting a discontinuous motion. Consequently, as the subspace dimension increases and the finer details are more prevalent, the ground truth lies further away from stable solutions leading to a higher possible reconstruction error. We will include results for higher dimensions. Note, however, that the subspace dimension cannot exceed the number of frames used for training. This is due to the SVD decomposition (used for dimensionality reduction) that is required for the WLS algorithm. Further, the computational time for CG and WLS is prohibitively high for online applications. We provide an example in the Table.

|  | SOC | CG | WLS |
|---|---|---|---|
|  | steam (r = 80) | | |
| $e(\hat{A})$ | **6.38** | 25.21 | 10.98 |
| time (s) | **5.45** | 1146.23 | 456.11 |

We will include results for the coffee cup sequence and also modify the transition to Section 2.1 (e.g., by mentioning that the literature on LDS has traditionally considered only no-input systems, which we will review first.)

**Reviewer 4**: Learning stable LDS will be most useful to systems that are known a priori to be stable. In practice, many systems have unforced dynamics that are naturally stable (asymptotically or in the sense of Lyapunov), such as mechanical systems, fluids, and electrical systems.

The novelty of this work lies in ensuring stability from the **first** iteration step of the optimization while achieving superior error performance and a lower space complexity. On the other hand, CG and WLS start from the LS solution and achieve stability only after converging. This difference can become crucial in online applications and time-sensitive tasks.

We will add references, among others, to stability preserving subspace identification methods (SPSIM) (i.e., Chui et al. (1996), Van Gestel et al. (2001), and Miller et al. (2013)). We omitted a comparison to such methods, given that WLS (2016) and CG (2007) compared and outperformed the seminal work in SPSIM by Lacy and Bernstein (2003) in terms of error, scalability, and execution time.

**Reviewer 5**: The product formulation of a matrix in SOC form bounds the maximum eigenvalues by constraining the $B$ matrix to have norm at most one. The detailed proof of the expression can be found in the original paper. The proposed algorithm enforces stability on the learned matrix. It does not rely on any assumptions (contrary to WLS that assumes time continuity of data) and is applicable to any task that seeks a stable matrix that maps data from $X$ to $Y$.

[Meta-Review · NeurIPS 2020]

The paper proposes a new method for learning a stable linear dynamical system from data, based on a recent paper [22] that shows that a matrix is stable if and only if it can be written as a product involving positive definite and orthogonal matrices. The proposed algorithm uses O(n^2) space in contrast to O(n^4) space used by previous similar methods, where n is the state dimension. The authors show that the new algorithm gets lower reconstruction error compared to baselines. Reviewers recommend acceptance and weren't concerned that the paper relies heavily on [22]. I agree and I suggest it be accepted as a poster.